# Cornuside Is a Potential Agent against Alzheimer’s Disease via Orchestration of Reactive Astrocytes

**DOI:** 10.3390/nu14153179

**Published:** 2022-08-03

**Authors:** Jun-Zhuo Shi, Xiao-Ming Zheng, Yun-Feng Zhou, Lu-Yao Yun, Dong-Mei Luo, Jiao-Jiao Hao, Peng-Fei Liu, Wei-Ku Zhang, Jie-Kun Xu, Yi Yan, Xin-Mei Xie, Yang-Yang He, Xiao-Bin Pang

**Affiliations:** 1School of Pharmacy, Henan University, Kaifeng 475004, China; 18803837386@163.com (J.-Z.S.); 18623823396@163.com (X.-M.Z.); zyf@henu.edu.cn (Y.-F.Z.); 18337278804@163.com (L.-Y.Y.); 15803830327@163.com (D.-M.L.); hao15764337935@163.com (J.-J.H.); 13383806192@163.com (P.-F.L.); 2Institutes of Traditional Chinese Medicine, Henan University, Kaifeng 475004, China; 3Institute of Clinical Medical Sciences, Department of Pharmacy, China-Japan Friendship Hospital, Beijing 100029, China; cpuzwk@163.com; 4School of Chinese Materia Medica, Beijing University of Chinese Medicine, Beijing 100029, China; xjkbucm@163.com; 5Institute for Cardiovascular Prevention (IPEK), Ludwig-Maximilians-University Munich, 80336 Munich, Germany; yannie0928@163.com; 6DZHK (German Centre for Cardiovascular Research), Partner Site Munich Heart Alliance, 80802 Munich, Germany

**Keywords:** Alzheimer’s disease, cornuside, reactive astrocytes, oxidative stress, neuroinflammation

## Abstract

Cornuside is an iridoid glycoside from *Cornus officinalis*, with the activities of anti-inflammatory, antioxidant, anti-mitochondrial dysfunction, and neuroprotection. In the present research, a triple-transgenic mice model of AD (3 × Tg-AD) was used to explore the beneficial actions and potential mechanism of cornuside on the memory deficits. We found that cornuside prominently alleviated neuronal injuries, reduced amyloid plaque pathology, inhibited Tau phosphorylation, and repaired synaptic damage. Additionally, cornuside lowered the release of interleukin-1β (IL-1β), interleukin-6 (IL-6), tumor necrosis factor-α (TNF-α), and nitric oxide (NO), lowered the level of malondialdehyde (MDA), and increased the activity of superoxide dismutase (SOD) and the level of glutathione peroxidase (GSH-Px). Cornuside also significantly reduced the activation of astrocytes and modulated A1/A2 phenotypes by the AKT/Nrf2/NF-κB signaling pathway. We further confirmed that LY294002 and Nrf2 silencing could block the cornuside-mediated phenotypic switch of C6 cells induced by microglia conditioned medium (MCM) in response to lipopolysaccharide (LPS), which indicated that the effects of cornuside in astrocyte activation are dependent on AKT/Nrf2/NF-κB signaling. In conclusion, cornuside may regulate the phenotypic conversion of astrocytes, inhibit neuroinflammation and oxidative stress, improve synaptic plasticity, and alleviate cognitive impairment in mice through the AKT/Nrf2/NF-κB axis. Our present work provides an experimental foundation for further research and development of cornuside as a candidate drug for AD management.

## 1. Introduction

As a progressive condition of neurodegeneration featuring cognitive impairment, behavioral abnormalities, and neuropsychiatric indications [1,2], Alzheimer’s disease (AD) is the chief pathogenic factor of dementia, which has become one of the greatest burdens of the century. Currently, the number of people having AD is over 50 million worldwide and, by 2050, the number is estimated to become 100 million. People develop dementia every three seconds and the current annual cost of dementia of approximately 1 trillion dollars is estimated to double by 2030 [3]. Unfortunately, first-line drugs used to treat AD only partially improve symptoms and do not delay progression. Despite the accelerated approval by the Food and Drug Administration (FDA) for managing AD in 2021, aducanumab remains under debate [4]. Thorough curing is currently impossible for AD [5]. Therefore, we still have a long way to go to uncover the pathological mechanisms of AD and develop effective therapeutic strategies for AD.

In addition to pathological features such as extracellular aggregation of amyloid beta (Aβ), intracellular neurofibrillary tangles (NFTs) [6,7], loss of neurons and synapses [8], glial cell activation, inflammation and oxidative stress exert crucial effects on the AD pathogenesis as well [9,10,11].

As the activation of gliocytes is a typical pathological trait of AD, astrocytes (the richest cerebral neurogliocytes in mammalians) are reported to regulate neuroinflammation and neurodegeneration in the central nervous system [12]. Emerging evidence shows that astrocytes provide metabolic substrates and antioxidant precursors to neurons [13], and regulate synaptic neurotransmission, synaptogenesis, and neurovascular coupling [14,15]. Of note, reactive astrocytes exert neurotoxic (A1) or neuroprotective (A2) phenotypes when stimulated by varying neuropathological factors, which resemble the M1/M2 phenotype of microglia/macrophages [16]. The A1 astrocytes, which are elicited by neuroinflammation, are capable of causing neuronal death, oligodendrocytic apoptosis, and synaptic disruption. Conversely, A2 astrocytes produce neuroprotective actions on neuron survival and synaptic repair via the upregulation of many neurotrophic factors and can be activated by ischemia [17]. A1 and A2 astrocytes have various gene profilings, which may account for the different roles they play in disease or injury [18,19]. Therefore, drugs that target reactive astrocytes activation against its toxicity while maintaining the neuroprotective effect would be potentially applicable for the AD management [20,21,22,23,24].

In East Asia, especially in Japan, Korea, and China, the use of Corni Fructus (dried *Cornus officinalis* Sieb. et Zucc fruits) [25] for edible and medicinal purposes has thousands of years of history. The chemical constituents of Corni Fructus include polysaccharides, iridoid glycosides, organic acids, tannins, esters, as well as aglycone [26]. The Crude Corni Fructus extract has been demonstrated to hold multiple pharmacological actions such as regulating immunity, anti-oxidation, hypoglycemic, anti-inflammation, anti-menopausal, neuroprotection, and bacteriostasis, which have thus been applied extensively in medicine and wellness product manufacturing [27,28,29,30,31,32,33]. Cornuside (7-*O*-Galloylsecologanol) is one of the major iridoid glycosides in *Cornus officinalis* (molecular structure is shown in Appendix A). According to extant reports, cornuside inhibited the expressions of cytokine-elicited adhesion and proinflammatory molecules in the endotheliocytes of humans [34], prohibited the carbon tetrachloride-triggered acute injury of the liver, and subdued the lipopolysaccharide-elicited inflammatory mediators through the NF-kB initiation repression in the RAW264.7 cells [35]. In addition, through the hinderance of histamine secretion and expression of proinflammatory cytokines, cornuside exerts a suppressive activity against the mastocyte-derived allergic and inflammatory reactions [36]. However, there are only a few studies reported about the pharmacological action of cornuside on neuroprotective activity. The studies reported that cornuside has a protective effect on cortical neuron damage after focal cerebral ischemia via the modulation of antioxidants and improved mitochondrial energy metabolism in rats [37]. In addition, cornuside could suppress BACE1, AChE, and BChE prominently, implying its therapeutic benefits for the AD comorbidities [38].

To date, there has been no evidence on the neuroprotective effects of cornuside through modulating the astrocyte phenotype in the 3 × Tg-AD (APP/PS1/Tau triple-transgenic model of AD) mice. Hence, the purpose of the present work is to explore cornuside’s action on the cognitive functionality among the 3 × Tg-AD mice, and its effects on astrocyte activation and phenotype switch, neuroinflammation, and oxidative stress as well. We found that cornuside could strengthen the murine learning and memory capacities, and raise the synaptic plasticity, which were partly attributed to the regulation of cornuside in modulating astrocyte phenotype through the AKT/Nrf2/NF-κB pathway and inhibiting neuroinflammation and oxidative stress.

## 2. Materials and Methods

### 2.1. Chemicals and Reagents

The supplier of cornuside (purity ≥ 98%), i.e., C_24_H_30_O_14_ (molecular mass: 542.9), was the Clinical Medicine Research Institute of China-Japan Friendship Hospital. The ELISA (enzyme-linked immunosorbent assay) kits for Aβ_1–40_ (E-EL-M3009), Aβ_1–42_ (E-EL-M3010), IL-1β (E-EL-M0037c), IL-6 (E-EL-M2453c), and TNF-α (E-EL-M3063) in mice, as well as for IL-1β (E-EL-R0012c), IL-6 (E-EL-R0896c), and TNF-α (E-EL-R2856c) in rats were products of Elabscience Biotechnology (Wuhan, China). Primary antibodies including anti-Tau (phospho Ser396) (ab32057), anti-GFAP (60190-1-Ig), anti-C3 (21337-1-AP), anti-S100A10 (11250-1-AP), anti-iNOS (18985-1-AP), anti-PSD95 (20665-1-AP), anti-Nrf2 (16396-1-AP), anti-Histone-H3 (17168-1-AP), anti-GAPDH (60004-1-Ig), and anti-β-actin (60004-1-Ig) antibodies were provided by Proteintech (Wuhan, China). Anti-Iba1 (019-19741) antibody was obtained from FUJIFILM Wako Pure Chemical Corporation. Anti-SYP (PB0462) was a product of Boster Biological Technology (Wuhan, China). The provider of anti-p-NF-κB p65 (#3033), anti-p-AKT (Ser473) (#4060), anti-AKT (#4691), anti-IκB-α(#4814), and anti-NF-κB p65 (#8242) antibodies was Cell Signaling Technology (Danvers, MA, USA), which was also the provider of both the goat anti-rabbit (#7074) and goat anti-mouse (#7076) secondary antibodies.

### 2.2. Animals

The supplier of 8-month-old 3 × Tg-AD mice was the Academy of Military Medical Sciences of China’s Laboratory Animal Center (Beijing, China). Meanwhile, the WT (wild-type, C57BL/6J) mice were procured from Sibeifu Biotechnology (Beijing, China). For the experimental cohort generation, the mice were reared in-house.

The mice were maintained at the ambient temperature of 25 °C and the relative humidity was controlled at 55%. The mice were exposed to alternating light for 12 h, and they could obtain food and water freely. The experiment was carried out in accordance with the Guideline for Care and Use of Laboratory Animals published by the US National Institutes of Health, and the Guidelines for the ethical review of laboratory animal welfare People’s Republic of China National Standard GB/T 35892-2018 [39], and it was approved by the Ethics and Animal Care and Use Committee of Henan University (approval No. HUSOM-2020-389). 

### 2.3. Experimental Design

Mice were randomized into 4 groups, namely the WT + Veh, 3 × Tg-AD + Veh, 3 × Tg-AD + cornuside (10 mg/kg, L), and 3 × Tg-AD + cornuside (20 mg/kg, H) groups. Intraperitoneal administration was given on the mice with cornuside solution in <1% DMSO or vehicle once a day for 4 weeks.

### 2.4. Morris Water Maze Test (MWM)

Experiments were carried out using a 120 cm diameter circular pool with a height of 40 cm, which was filled with water (22 ± 2 °C, opaque property due to the addition of nontoxic white dye) and consisted of 4 identical quadrants. At one quadrant’s center, a platform 8 cm in diameter was arranged 1 cm below the surface of water. There were a variety of special markings in the pool, which served as spatial references. The mice were then tested three times daily for 5 successive days at 30 min intervals between sessions, and the escape incubation period was recorded. Once the mice found the platform, it would stay thereon for a minimum of 5 s. For any mice incapable of finding the platform within the designated time, guidance was given to reach and maintain thereon for 10 s. The platform was withdrawn from the pool on the 6th d for probe testing to test memory consolidation. The cut-off time for probe test was 90 s. The time and distance spent by each mouse in the target quadrant were recorded [40]. Data documentation was accomplished with the ZS-001 analyzer (Beijing, China).

### 2.5. Nissl Staining

Nissl staining was applied to observe neuronal morphologic changes. The mice were anesthetized; they were first perfused with saline and then fixed with 4% paraformaldehyde. The brain tissues were cut into coronal sections, embedded in paraffin, and further cut into slices with 5 μm thickness. The slices were baked in a thermostat at 80 °C for 1 h, deparaffinized in xylene, and then dehydrated in a series of ethanol solutions with decreasing concentration (100%, 85%, and 75%). Afterward, tissue sections were stained with Nissl solution (#C0117, Beyotime Biotechnology, Shanghai, China) for 8 min, rinsed in ultrapure water, dehydrated in 100% alcohol (I, II) for 2 min each, cleared in xylene for 3 min, and mounted [41]. Images of brain sections on cover glass were obtained by fluorescence microscopy (Olympus IX53, Tokyo, Japan).

### 2.6. Thioflavin S-Staining

The 5 μm brain slices were incubated with 0.3% potassium permanganate solution for 5 min, 1% oxalic acid solution for 5 min, and 1% sodium borohydride solution for 5 min. After phosphate-buffered saline (PBS) rinses twice, the slices were stained with 0.5% Thioflavin S solution (dissolved in distilled water containing 50% ethanol) for 15 min, and then incubated with 10 μg/mL of DAPI for 10 min. Afterward, the sections were dehydrated with a series of ethanol solutions with increasing concentration (70%, 80%, 90%, and 100% ethanol) for 2 min, and sealed with an anti-fluorescence attenuation sealing sheet (Solarbio, Beijing, China). Fluorescence imaging was collected using an Olympus fluorescence microscope (Olympus IX53, Tokyo, Japan).

### 2.7. Immunohistochemical Staining

The 5 μm brain slices were dewaxed, rehydrated, and incubated in 3% H_2_O_2_ (*w*/*v*) for 10 min and 0.3% triton X-100 in PBS at room temperature for 10 min. Then, the antigens in slices were retrieved with Tris-EDTA repair solution and blocked with 5% BSA. Mouse brain sections were incubated with primary antibodies at 4 °C overnight. Finally, stainings were visualized with secondary antibodies. Images were acquired using an Olympus fluorescence microscope. The same image acquisition settings were used for each staining.

### 2.8. Immunofluorescence Double Staining

Immunofluorescence double staining was performed on brain tissue samples. Paraffin sections of 5 μm were deparaffinized by xylene and a series of ethanol solutions with decreasing concentration (100%, 95%, 90%, 85%, 80%, 75%, and 70% ethanol), then the antigens were retrieved with Tris-EDTA repair solution and blocked with 5% BSA. After that, astrocyte (A1) was stained with anti-C3 antibody and anti-GFAP antibody in a humidified chamber at 4 °C overnight. After three washes in PBS, slices were incubated with goat anti-mouse IgG H&L (Alexa Fluor 555, 1:100) and goat anti-rabbit IgG H&L (Alexa Fluor 488, 1:100) for 1 h at room temperature, followed by PBS wash, DAPI (10 μg/mL) incubation in the dark for 10 min, and sealing with an anti-fluorescence attenuation sealing sheet. Images were obtained using an Olympus fluorescence microscope (Olympus IX53, Tokyo, Japan).

### 2.9. Cell Culture

C6 cells were purchased from Saibai Kang Biotechnology Co., Ltd. (Shanghai, China) and cultured in F12k medium with 15% horse serum, 2.5% fetal bovine serum (FBS; Zeta Life, Inc., San Francisco, CA, USA), 100 U/mL of penicillin, and 100 U/mL of streptomycin at 37 °C in a humidified atmosphere of 5% CO_2_. BV2 cells were purchased from Saibai Kang Biotechnology Co., Ltd. (Shanghai, China) and cultured in Dulbecco’s modified Eagle’s medium (DMEM; Thermo Fisher Scientific, Inc., Waltham, MA, USA) supplemented with 10% FBS, 100 U/mL of penicillin, and 100 U/mL of streptomycin at 37 °C and 5% CO_2_.

### 2.10. Preparation of MCM

After a 24 h incubation of BV2 cells cultured in serum-free DMEM filled with 2 μg/mL of lipopolysaccharide (LPS), the microglia-conditioned medium (MCM) was acquired. Meanwhile, an LPS-free medium conditioned by microglia was used as a control medium for MCM herein.

### 2.11. Real-Time Quantitative PCR

The real-time quantitative PCR (RT-qPCR) assay was carried out in the present study in line with previously reported methods [42,43]. Cerebral tissue sampling of total RNA was accomplished with TRIzol (Invitrogen, San Francisco, CA, USA) as per the guidelines of the manufacturer. The RNA isolates were quantified utilizing a Nanodrop-2000 spectrophotometer. For the synthesis of cDNA templates from the total RNA, a Transcriptor First-Strand cDNA Synthesis kit (Vazyme RT kit, Nanjing, China) was utilized. A Real-Time PCR System (StepOnePlus; Vazyme, Nanjing, China) was exploited for the q-PCR assessment using the Master Mix (ChamQ Universal SYBR). Normalization of target gene levels was accomplished against GAPDH. The mRNA expressions of genes were assessed by the 2^−^^△△Ct^ approach. Appendix A shows the details of the q-PCR primer sequences.

### 2.12. ELISA

The supernatant and cerebral tissue expressions of such inflammatory cytokines as interleukin-1β (IL-1β), interleukin-6 (IL-6), and tumor necrosis factor-α (TNF-α) were examined via ELISA kits as per the guidelines of the manufacturer (Elabscience, Wuhan, China). Analysis of 450 nm absorbance was accomplished with the aid of the microplate reader.

### 2.13. NO Testing

The murine cortex and hippocampus were pulverized using PBS, and then 10% homogenates were taken and subjected to a 10 min centrifugation under 4 °C and 2000× *g* conditions. Supernatant gathering from cells treated with MCM was accomplished either with or without cornuside. Thereafter, the supernatant and homogenate contents of Nitric Oxide (NO) were examined as per the assay kit (Jiancheng Bioengineering Institute, Nanjing, China) protocol (Nitrate reductase method).

### 2.14. Measurement of MDA, SOD, and GSH-Px

The hippocampus and cortex of mice in each group were weighed, homogenized, and then subjected to a 15 min centrifugation (3000× *g*) for the tissue supernatant extraction. Acquisition of cellular supernatant was accomplished from the MCM-induced cells either with or without cornuside. For the malondialdehyde (MDA) and glutathione peroxidase (GSH-Px) level and the superoxide dismutase (SOD) activity assessment, the MDA assay kit (TBA method; Jiancheng Bioengineering Institute, Nanjing, China), the GSH-Px colorimetric assay kit (Elabscience, Wuhan, China), and the T-SOD (total SOD) colorimetric assay kit (WST-1 method) were separately utilized as per the protocols of manufacturers.

### 2.15. Western Blot

Western blot was conducted following the prior procedure [44,45]. The BCA protein assay kit (Solarbio Science & Technology, Beijing, China.) was utilized for quantifying the protein contents. Isolation of the protein extracts was accomplished via SDS-PAGE, followed by shifting onto the PVDF membranes (0.22 μm) for 70 min. A 2 h blockage of the membranes proceeded with the use of 5% non-fat milk in TBST (Tris-buffered saline involving 0.1% Tween 20), and subsequently, the membranes were subjected to an overnight incubation at 4 °C using the primary antibodies shown below: Anti-Tau (phospho Ser396), anti-Iba1, anti-GFAP, anti-C3, anti-S100A10, anti-iNOS, anti-PSD95, anti-SYP, anti-BDNF, anti-AKT, anti-NF-κB p65, anti-p-AKT, anti-Nrf2, anti-p-NF-κB p65, or anti-IκB-α antibody. After thrice TBST washing, an additional 2 h incubation of the membranes was accomplished using a goat anti-rabbit or goat anti-mouse secondary antibody. For the collection and analysis of images, Image J 2.0 was utilized.

### 2.16. Statistical Analysis

Data were statistically processed by GraphPad Prism 8 and Origin 8.0, and expressed as means ± SEM. One-way ANOVA was employed to make comparisons among groups, and in the case of the normal distribution, Tukey’s post hoc test was conducted. Otherwise, the Kruskal–Wallis test was adopted. Differences were regarded as significant with *p*  <  0.05.

## 3. Results

### 3.1. Cornuside Ameliorated Cognitive Deficits in 3 × Tg-AD Mice

The Morris water maze test showed that 3 × Tg-AD mice exhibited significant cognitive dysfunction, including spatial and nonspatial cognitive impairments. In the training phase, mice in the 3 × Tg-AD + Veh group showed a notable increase in escape latency on the hidden platform test compared with those in the WT + Veh group, while 3 × Tg-AD mice receiving cornuside treatment showed a shorter incubation period and escape distance, which was significantly different from those of the transgenic mice group (Figure 1a,b).

When the test platform was hidden, the 3 × Tg-AD + Veh mice exhibited an aimless search strategy, while the mice with cornuside treatment spent a larger proportion of time and distance in the target quadrant, and the number of times it crossed the platform position was similar to the WT + Veh group (Figure 1c–f). Taken together, these results showed that cornuside (20 mg/kg) attenuated the cognitive impairments in 3 × Tg-AD mice, which was used for the following experiments.

### 3.2. Cornuside Improved the Histomorphology, Reduced Amyloid Plaque and Inhibited Tau Phosphorylation in Brains of 3 × Tg-AD Mice

With the purpose of determining whether cornuside affected the pathological alterations in AD, the dense Aβ plaque counts, Tau phosphorylation, and histopathological changes were evaluated in the case of 3 × Tg-AD mice. Thioflavin S-staining showed that the hippocampal burden of Aβ plaques was mitigated by cornuside among 3 × Tg-AD mice (Figure 2a,b). Markedly declined Aβ_1-40_ and Aβ_1-42_ in the hippocampus and cortex were noted in these mice (Figure 2c,d). Furthermore, the results showed that cornuside treatment significantly inhibited the phosphorylation of Tau at the site of Ser396 in 3 × Tg-AD mice (Figure 2e,f). In addition, as revealed by Nissl’s staining, no Nissl substance was present in the murine neuronal cells, while the pathological neuronal degradation was improved by cornuside in the 3 × Tg-AD mouse brains (Figure 2g).

### 3.3. Cornuside Inhibited the Activation of Glial Cells in 3 × Tg-AD Mice

We first measured the astrocyte-specific gene markers at mRNA levels. As expected, astrocyte-specific markers GFAP and AQP4 were higher in 3 × Tg-AD mice compared to WT + Veh mice, which could be attenuated by cornuside treatment, suggesting a suppressive role of cornuside in the activation of astrocytes in 3 × Tg-AD mice (Figure 3a).

Next, the expression of GFAP was detected at the protein level to investigate the effect of cornuside on astrocytes in 3 × Tg-AD mice. The immunohistochemical results showed that the expression of GFAP was increased significantly in 3 × Tg-AD mice, and a significant reduction was observed in cornuside-treated mice (Figure 3b,c). We also examined the expression of Iba1, the signature protein of microglia. Both GFAP and Iba1 were significantly upregulated in the cortex and the hippocampus of 3 × Tg-AD mice compared with WT mice, implicating an overactivation of glial cells. The upregulation of GFAP and Iba1 expression were abolished in 3 × Tg-AD mice with the treatment of cornuside (Figure 3d–f).

### 3.4. Cornuside Regulated A1/A2 Astrocytic Phenotype Alteration in 3 × Tg-AD Mice

To clarify whether cornuside modifies A1/A2 astrocytic phenotype alteration, we next determined the expression of selected A1- and A2-specific transcription factors by quantitative PCR in the cortex. It showed that the expressions of the A1 astrocyte markers psmb8, Serping1, C3, and Amigo2 were significantly upregulated except for Gbp2 in 3 × Tg-AD mice, and all these markers were reduced after cornuside treatment (Figure 4a). The A2 astrocyte markers Ptx3, S100A10, Ptgs2, Tgm1, and CD14 were downregulated in 3 × Tg-AD mice, which could be restored after cornuside treatment (Figure 4b). We further measured the protein expression of the A1-type astrocyte marker C3 and the A2-type astrocyte marker S100A10 in the cortex and hippocampus. Western blot analysis showed that the C3 protein level was markedly increased in the 3 × Tg-AD mice, and this effect was reduced by cornuside treatment (Figure 4c,d). The S100A10 protein level was markedly decreased in the 3 × Tg-AD mice, and this effect was reversed by cornuside treatment (Figure 4c,e). Immunofluorescence also confirmed that C3/GFAP-positive cells were significantly increased in the 3 × Tg-AD mice compared with WT mice, while these positive cells were significantly decreased in cornuside-treated mice (Figure 4f). These results indicated that cornuside induced the transition from the A1 phenotype to A2 phenotype in astrocytes.

### 3.5. Cornuside Reduced Neuroinflammation and Oxidative Stress in Brains of 3 × Tg-AD Mice

During the pathological process of AD, reactive astrocytes produce inflammatory cytokines and induce oxidative stress, which, in turn, exacerbate synaptic disorders, reduce neuronal health, and increase Aβ production. Therefore, we detected the levels of some inflammatory factors. The results showed that the levels of TNF-α, IL-1β, IL-6, NO, and iNOS were significantly increased in the cortex and hippocampus of 3 × Tg-AD mice, and the administration of cornuside could inhibit the increase in these inflammatory factors (Figure 5a–f). Accordingly, we measured the activity of SOD and the levels of GSH-Px and MDA. The results showed that the activity of SOD and the level of GSH-Px were significantly reduced and the level of MDA was markedly enhanced in 3 × Tg-AD mice compared with those of WT + Veh mice; however, cornuside markedly relieved the oxidative stress status (Figure 5g–i).

### 3.6. Cornuside Increased Neurotrophic Factors and Synapse-Associated Proteins in 3 × Tg-AD Mice

It is evident that A1 reactive astrocytes are synaptic, destructive, and neurotoxic, and A2 reactive astrocytes, in contrast, upregulate many neurotrophic factors promoting neuronal survival, as well as synaptic repair, suggesting that A2 may have a nutritional repair function [17]. Next, we measured the mRNA and protein levels of the neurotrophic factor in astrocytes to investigate whether cornuside has a neuroprotective effect to induce the phenotypic switch from A1 to A2. It was demonstrated that the protein level of BDNF and the mRNA levels of Thbs1, TGF-β, and BDNF were significantly decreased in 3 × Tg-AD mice, which could be restored in cornuside-treated mice (Figure 6a–c). The results indicated that the treatment of cornuside changed the neurotoxic A1 type into the neuroprotective A2 type, facilitated the release of nutritional factors, and had neurotrophic and neuroprotective effects in 3 × Tg-AD mice.

To investigate synaptic dysfunction in 3 × Tg-AD mice, we detected synaptic proteins including SYP and PSD95. The results showed that SYP and PSD95 were significantly decreased in the brain tissues of 3 × Tg-AD mice, while cornuside-treated mice exhibited a higher expression of SYP and PSD95 compared to that of 3 × Tg-AD mice (Figure 6a,d,e). These results indicated that the treatment of cornuside could repair synaptic damage.

### 3.7. Cornuside Activated AKT/Nrf2 Pathway and Inhibited NF-κB Signaling in 3 × Tg-AD Mice

To investigate the mechanism of A1/A2 astrocytic alteration regulated by cornuside, AKT/Nrf2/NF-κB signaling pathways were examined. We first investigated whether cornuside could affect the AKT/Nrf2 signaling pathway in 3 × Tg-AD mice. The expressions of p-AKT and nuclear Nrf2 were significantly decreased in the brain tissues of 3 × Tg-AD mice, while the expressions of these proteins were increased in cornuside-treated mice (Figure 7a–c). These data indicated that cornuside activated the AKT/Nrf2 pathway in 3 × Tg-AD mice. The activation of the NF-κB pathway could upregulate the production of pro-inflammatory cytokines, and IκB-α is reported to act as an inhibitor of NF-κB. Next, we examined the NF-κB signaling pathway, and it was demonstrated that the level of IκB-α was downregulated and p-NF-κB-p65 was upregulated in the cortex and hippocampus of 3 × Tg-AD mice, while treatment of cornuside could suppress the activation of the NF-κB pathway (Figure 7a,d,e). These results suggest that cornuside may affect phenotypic changes of astrocytes through the AKT/Nrf2/NF-κB signaling pathway.

### 3.8. Cornuside Regulated A1/A2 Phenotypic Switch of C6 Cells Induced by MCM

First, as determined by ELISA and NO testing, the pro-inflammatory cytokines (IL-1β, IL-6 and TNF-α) and NO production peaked in MCM of BV2 cells with LPS treatment at the concentration of 2 µg/mL (Appendix A). Next, we evaluated the effects of cornuside on cell viability in C6 cells. MTT results showed that cornuside ranging from 0 to 20 μmol/L had no obvious effect on the cell viability of C6 cells 24 h post-treatment (Appendix A). Cornuside (10 μmol/L) was then used to treat C6 cells in the presence or absence of MCM.

To clarify whether cornuside (10 μmol/L) modifies A1/A2 astrocytic alteration induced by MCM, the mRNA levels of C3 and S100A10 in C6 cells were examined. We found that MCM significantly increased C3 (A1 astrocyte marker) mRNA expression, which was inhibited by cornuside. In addition, MCM suppressed S100A10 (A2 astrocyte marker) mRNA expression in C6 cells, which was successfully rescued by cornuside (Figure 8a,b).

### 3.9. Cornuside Protected C6 Cells Induced by MCM via AKT/Nrf2/NF-κB

Next, we utilized PI3K inhibitor (LY294002) and Nrf2 siRNA in vitro to determine whether cornuside improved astrocyte’s phenotypic switch induced by MCM and whether it was dependent on the AKT/Nrf2/NF-κB signaling pathway. We then constructed LPS-induced MCM to simulate the AD extracellular microenvironment. First, the impact of LY294002 on the Nrf2 and NF-κB signaling pathways was investigated. Compared with the cornuside-treated group, LY294002 significantly attenuated cornuside-induced AKT activation, nuclear-Nrf2, and IκB-α activation and decreased cornuside-repressed p-NF-κB p65 activation (Figure 9a–e). These results suggested that Nrf2 and NF-κB signals are downstream of the AKT pathway mediated by cornuside. Second, we used Nrf2-silenced C6 cells to verify whether the effect of cornuside was mediated by Nrf2. The RNA interference efficiency was verified by Western blot analyses (Appendix A). Compared with the cornuside-treated group, we found that Nrf2 silencing did not result in any significant change in phosphorylation status of AKT (Figure 9f,g), suggesting that Nrf2 serves as the downstream effector of the AKT pathway following cornuside treatment. In addition, Nrf2 silencing significantly attenuated cornuside-induced IκB-α activation, and abolished the repression of p-NF-κB p65 activation in response to MCM by cornuside (Figure 9f,i,j). These results proved that both the inactivation of AKT and ablation of Nrf2 blocked the treatment effect of cornuside.

### 3.10. Effects of PI3K Inhibitor and Nrf2 Silencing on Astrocyte Function

Next, we verified the effects of PI3K inhibitor (LY294002) and Nrf2 silencing on MCM-induced C6 cells function. As expected, LY294002 and Nrf2 siRNA attenuated the cornuside-induced down-regulation of A1 marker C3 mRNA expression and up-regulation of A2 marker S100A10 mRNA expression (Figure 10a,b,j,k). Furthermore, LY294002 and Nrf2 knockdown significantly reversed the inhibitory effects of cornuside on IL-1β, IL-6, TNF-α, NO, and MDA production (Figure 10c–f,i,l–o,r) and the boost effects of cornuside on SOD and GSH-Px production (Figure 10g,h,p,q) in the presence of MCM. Thus, the results demonstrated that AKT/Nrf2/NF-κB was crucial to the regulation of astrocyte phenotypic changes mediated by cornuside, and LY294002 and Nrf2 silencing attenuated the antioxidant and anti-neuroinflammatory properties of cornuside.

## 4. Discussion

AD refers to an intricate and progressive condition of dysfunctional cognition, which features behavioral alterations, as well as cognitive defects [1]. Currently, no agent is capable of preventing the incidence and progression of AD given its complicated pathological traits and elusive pathogenesis, which is worthy of further study [46]. Cornuside is one of the major iridoid glycosides in *Cornus officinalis*, which can fight against inflammation and oxidation and protect the nervous system [26,27,30,34,35,36,37]. In this study with the 3 × Tg-AD mice, cornuside was found to ameliorate the neurological deficits remarkably, alleviate neuronal injuries, reduce amyloid plaque pathology, inhibit Tau phosphorylation, and repair synaptic damage. In addition, it also relieved the neuroinflammation and oxidative stress in brain tissues of 3 × Tg-AD mice via reducing levels of IL-1β, IL-6, TNF-α, NO, and MDA and increasing the GSH-Px level and the SOD activity. We also found that cornuside significantly reduced the astrocytic and microglial activation, and coordinated the A1/A2 phenotypes of astrocytes by up-regulating the AKT/Nrf2 pathway and down-regulating the NF-κB pathway.

Astrocyte and microglia are important immune cells in AD brains, which modulate neuroinflammation, oxidative stress, and neurodegeneration in the CNS [46]. Emerging evidence suggests that the resident microglia are transformed into proliferative and pro-inflammatory microglia, which increase the capacity to induce astrocyte reactivity, mainly composed of reactive astrocytes. Recent studies have also shown that reactive astrocytes have two distinct phenotypes, A1 and A2, which play neurotoxic and neuroprotective roles, respectively [47,48,49]. These activated astrocytes are potential therapeutic targets for AD. In the present study, GFAP and Iba1 were significantly upregulated in the cortex and the hippocampus of 3 × Tg-AD mice, a hallmark of overactivation of glial cells, which could be reversed by cornuside. The increased C3 (A1 astrocyte marker) and decreased S100A10 (A2 astrocyte marker) in 3 × Tg-AD mice were also abrogated by cornuside. These results suggested a detrimental role of A1 astrocytes and a beneficial role of A2 astrocytes in the development of AD. Regulation of the A1/A2 astrocyte ratio may be a new strategy to prevent AD.

Neuroinflammation and oxidative stress induced by activated astrocytes contribute substantially to AD [50,51]. Astrocytes are excessively activated, and A1 astrocytes release robust levels of inflammatory cytokines, such as IL-1β, IL-6, and TNF-α, which contribute to severe inflammatory responses and affect synaptic function [14,15,16]. Meanwhile, during AD disease progression, the interaction between oxidative stress and inflammation can lead to a vicious cycle to aggravate the neural damage. Reactive astrocytes produce inflammatory cytokines and fuel the oxidative stress in AD, which, in turn, aggravate synaptic disturbances, reduce neuronal health, and enhance Aβ production. In contrast, A2 reactive astrocytes up-regulate many neurotrophic factors that promote neuron survival and growth, as well as the platelet reactive protein that promotes synaptic repair, suggesting that A2 may have a nutritional repair function [14,15,52,53]. Our study confirmed that the levels of TNF-α, IL-1β, IL-6, and NO and the content of MDA were significantly enhanced, and the activity of SOD and the level of GSH-Px were greatly reduced in 3 × Tg-AD mice. Meanwhile, cornuside effectively inhibited the inflammatory response and relieved the oxidative stress status in the model mice. Meanwhile, the levels of neurotrophic factors (BDNF, Thbs1, and TGF-β) were significantly decreased in 3 × Tg-AD mice, which could be restored in cornuside-treated mice. The results indicated that the treatment of cornuside facilitated the phenotype switch from the neurotoxic A1 to the neuroprotective A2 by manipulation of nutritional factors.

PI3K/AKT pathway activation occurs in response to a variety of extracellular signals, and it inhibits neuroinflammation and protects the nervous system; for instance, it regulates the secretion of TGF-β, a beneficial factor for the brain [54]. NF-κB is a key nuclear transcription factor that plays an important role in the regulation of inflammation and immune responses [55,56]. NF-κB exists in the cytoplasm mainly in an inactive form and is regulated by NF-κB inhibitor (IκB). Once activated, NF-κB intensifies and amplifies inflammatory responses by increasing the production of pro-inflammatory cytokines [57,58]. Previous studies have shown that regulation of the PI3K/AKT and NF-κB signaling pathway can regulate A1/A2 phenotypic switches in astrocytes [20,59]. Li et al. found that microglia induced the transformation of astrocytes to the A1 phenotype in the spinal cord by reducing the activation of the CXCR7/PI3K/AKT signaling pathway during CPSP [59]. In addition, it has been reported that milk fat globule epidermal growth factor 8 can decrease the expression of A1 astrocytes and increase the expression of A2 astrocytes by upregulating the activation of the PI3K/AKT pathway and downregulating the activation of the NF-κB cell pathway [20]. In line with the previous studies, the downregulated AKT pathway and upregulated NF-κB signaling were documented in brain tissues of 3 × Tg-AD mice, which could be reversed by cornuside via regulation of A1/A2 astrocytic phenotype alteration in 3 × Tg-AD mice. Furthermore, reactive astrocytes induce neuroinflammation, which, in turn, exacerbates oxidative stress in the brain. The Nrf2 signaling pathway plays a pivotal role in regulation of adaptive oxidative stress and neuroinflammation, and Nrf2 is evident in preventing oxidative stress-induced cell damage [60,61]. Accumulating experimental evidence has also demonstrated that protein kinase B (AKT) plays an important role in the nuclear localization of Nrf2 [62,63,64]. In support of this notion, we found that nuclear-Nrf2 expression was clearly declined in 3 × Tg-AD mice, which could be rescued in cornuside-treated mice. To verify the molecular mechanism of cornuside in regulating the transformation of reactive astrocytes, LY294002 and Nrf2 silencing were demonstrated to abolish the induction of AKT activation, localization of nuclear Nrf2, and activation of IκB-α, as well as the reduction in p-NF-κB p65 mediated by cornuside. The PI3K inhibitor and Nrf2 silencing also blocked the antioxidant and anti-neuroinflammatory properties of cornuside in C6 cells. All these findings suggest that cornuside-mediated transformation of reactive astrocytes in AD is in an AKT/Nrf2/NF-κB-dependent manner.

## 5. Conclusions

In conclusion, our results indicated that cornuside could regulate the phenotypic conversion of astrocytes, inhibit neuroinflammation and oxidative stress, improve synaptic plasticity, and alleviate cognitive impairment in mice through the AKT/Nrf2/NF-κB axis (Figure 11). Therefore, cornuside might be a promising drug for AD therapy, although extensive efforts are still needed.

## Figures and Tables

**Figure 1 nutrients-14-03179-f001:**
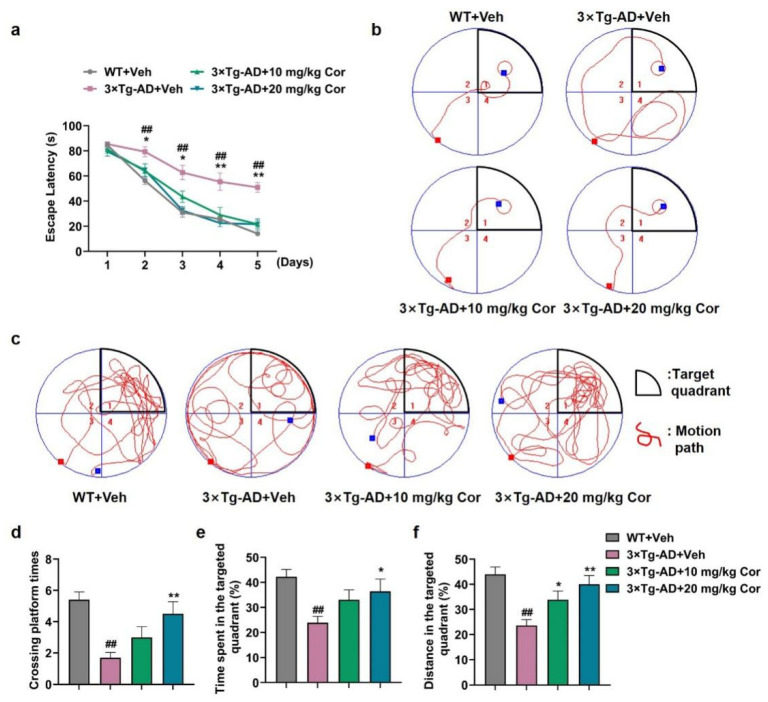
Cornuside alleviated the learning and memory deficits among 3 × Tg-AD mice. Training trials were implemented 4 times daily for a total of 5 successive days. The distance and duration of swimming prior to the platform arrival were documented in an automatic manner. Twenty-four hours after the trials, probe tests were conducted. (**a**) The escape latency. (**b**) The representative track in the probe test on the 5th day for the mice. (**c**) The representative track in the platform-free probe test on the 6th day for the mice. (**d**) The number of platform crossings in the probe test. (**e**) Time consumption within the target quadrant. (**f**) Overall distance of swimming within the targeted quadrant. Data are all denoted as mean ± SEM (*n* = 10 in each group). ^##^ *p* < 0.01 vs. WT + Veh group; * *p* < 0.05 and ** *p* < 0.01 vs. 3 × Tg-AD + Veh group.

**Figure 2 nutrients-14-03179-f002:**
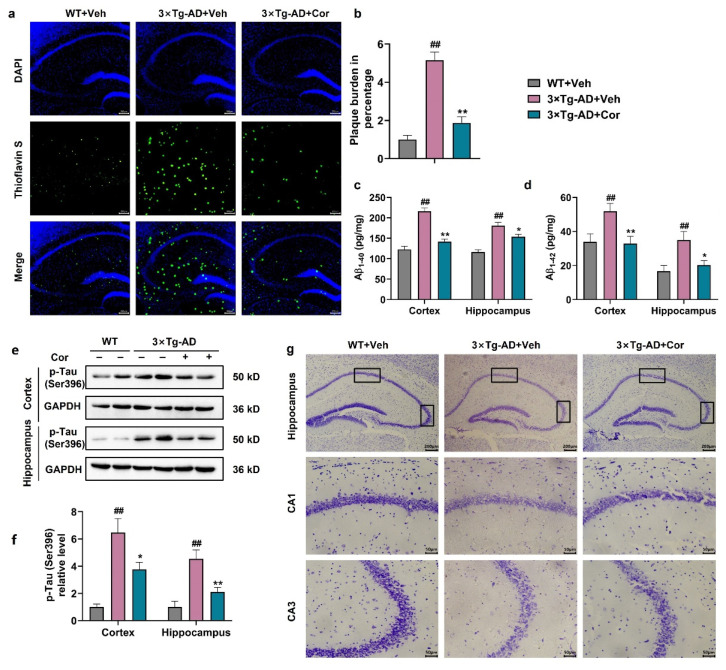
Cornuside improved the histomorphology, reduced amyloid plaque and inhibited Tau phosphorylation in the 3 × Tg-AD mouse brains. (**a**) Representative Thioflavin S immunofluorescent images of Aβ plaque-occupied hippocampal region for the 3 × Tg-AD + Veh, 3 × Tg-AD + cornuside, and WT + Veh mice (*n* = 4 in each group). Scale bar for original micrographs = 200 μm. (**b**) Surface area quantification for the Aβ plaques. (**c**,**d**) The level of Aβ_1-40_ (**c**) and Aβ_1-42_ (**d**) in each experimental group (*n* = 5 per group). (**e**) The representative images and (**f**) quantification analysis of p-Tau (Ser396) protein in the hippocampi and cortices of 3 × Tg-AD mice with or without cornuside treatment compared with WT control mice (*n* = 6 per group). (**g**) Nissl’s staining of the neuronal cells and typical micrographs from every experimental group. Scale bar for original micrographs = 200 μm, whereas that for enlarged graphs = 50 μm. Data are all expressed as mean ± SEM. ^##^ *p* < 0.01 compared with WT + Veh; * *p* < 0.05 and ** *p* < 0.01 compared with 3 × Tg-AD mice + Veh.

**Figure 3 nutrients-14-03179-f003:**
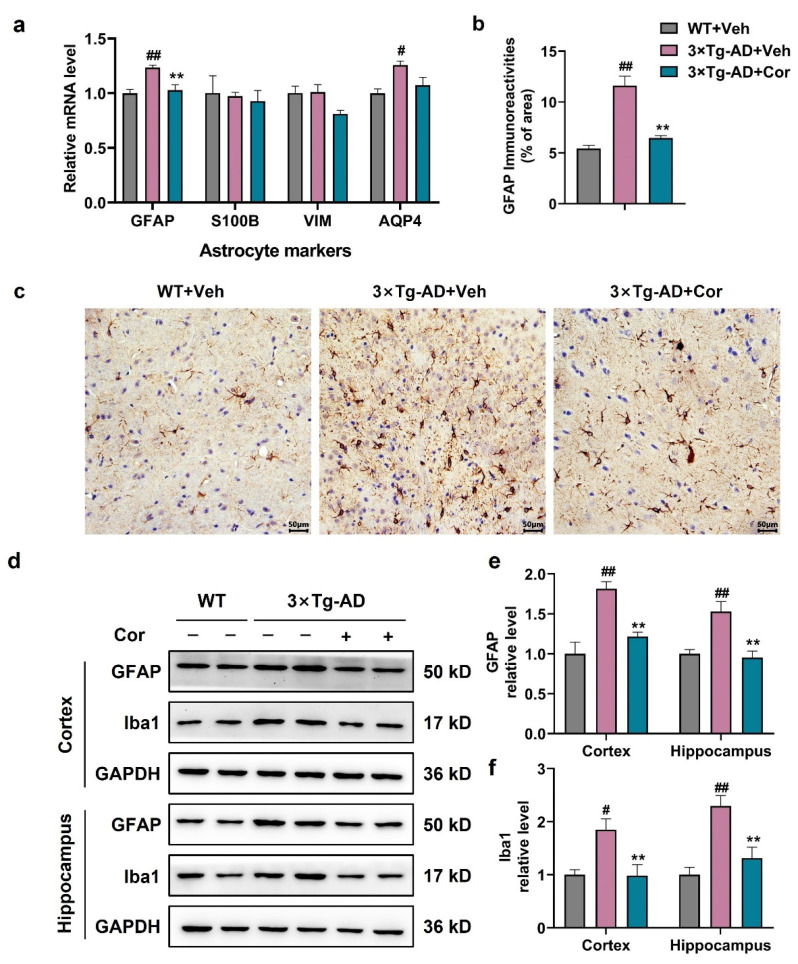
Cornuside inhibited glial activation in 3 × Tg-AD mice. (**a**) Quantitative PCR of astrocyte markers GFAP, S100B, Vim, and AQP4 from the cortex of 3 × Tg-AD mice in the presence or absence of cornuside compared to WT control mice. The reference gene used was GAPDH, *n* = 5 per group. (**b**) Quantification and (**c**) immunohistochemical images of GFAP-positive area in the cortices of experimental mouse brains. (**d**) The typical WB images and (**e**,**f**) quantitative assessments of GFAP and Iba1 protein in the cortex and hippocampus of 3 × Tg-AD mice in the presence or absence of cornuside compared with WT control mice (*n* = 6 per group), respectively. Data are shown as mean ± SEM. ^#^ *p* < 0.05 and ^##^ *p* < 0.01 relative to WT + Veh; ** *p* < 0.01 relative to 3 × Tg-AD + Veh group.

**Figure 4 nutrients-14-03179-f004:**
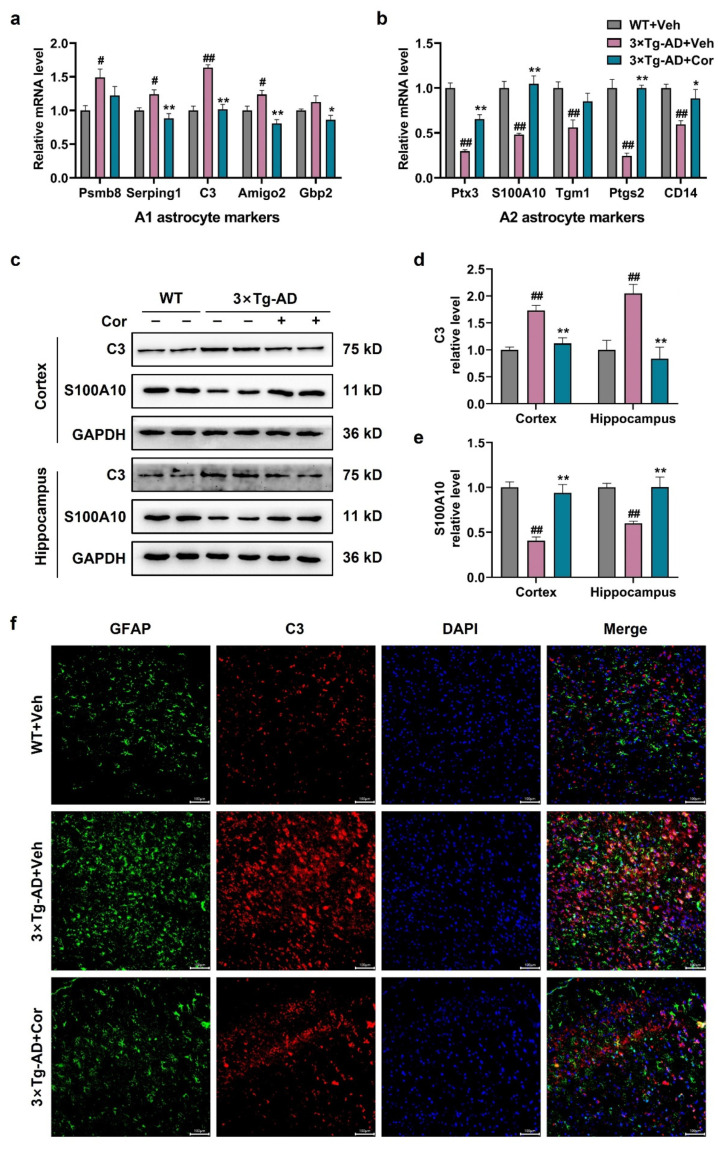
Cornuside-regulated A1/A2 astrocytic phenotype alteration in 3 × Tg-AD mice. (**a**) The mRNA expression of the A1 astrocyte markers Psmb8, Serping1, C3, Amigo2, and Gbp2 in 3 × Tg-AD mice in the presence or absence of cornuside compared to WT control mice (*n* = 5 per group). (**b**) The mRNA expression of the A2 astrocyte markers Ptx3, S100A10, Tgm1, Ptgs2, and CD14 in 3 × Tg-AD mice in the presence or absence of cornuside compared to WT control mice (*n* = 5 per group). (**c**) The representative images and (**d**,**e**) quantification analysis of C3 protein (A1 astrocyte marker) and S100A10 protein (A2 astrocyte marker) in the hippocampi and cortices of 3 × Tg-AD mice with or without cornuside treatment compared with WT control mice (*n* = 6 per group). (**f**) Representative immunofluorescent micrographs indicating the C3 and GFAP co-localization in astrocytes in the cortex. Scale bar = 100 μm. Data presented are mean ± SEM. ^#^
*p* < 0.05 and ^##^
*p* < 0.01 compared with WT + Veh; * *p* < 0.05 and ** *p* < 0.01 compared with 3 × Tg-AD + Veh group.

**Figure 5 nutrients-14-03179-f005:**
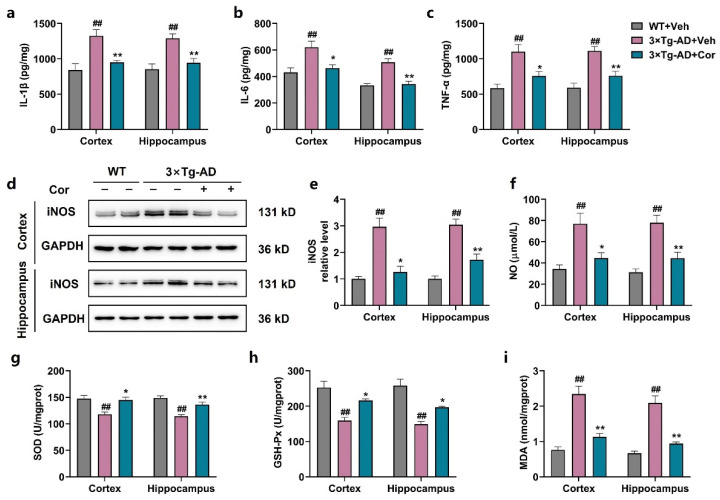
Cornuside reduced neuroinflammation and oxidative stress in the brains of 3 × Tg-AD mice. (**a**–**c**) Inflammatory cytokines (IL-6, IL-1β, and TNF-α) in the cortex and hippocampus of cornuside or vehicle-treated 3 × Tg-AD mice versus WT controls (*n* = 5 per group). (**d**) Typical micrographs and (**e**) quantitative assessments of hippocampal and cortical iNOS protein for the 3 × Tg-AD mice administered with cornuside/vehicle versus the WT controls (*n* = 6 per group). (**f**) The content of NO in the cortex and hippocampus of cornuside or vehicle-treated 3 × Tg-AD mice versus WT controls (*n* = 5 per group). (**g**–**i**) The GSH-Px, SOD, and MDA expressions in the cortex and hippocampus of cornuside or vehicle-treated 3 × Tg-AD mice in contrast to the WT controls (*n* = 5 per group). Values are all expressed as mean ± SEM. ^##^ *p* < 0.01 compared with WT-Con; * *p* < 0.05 and ** *p* < 0.01 compared with 3 × Tg-AD group.

**Figure 6 nutrients-14-03179-f006:**
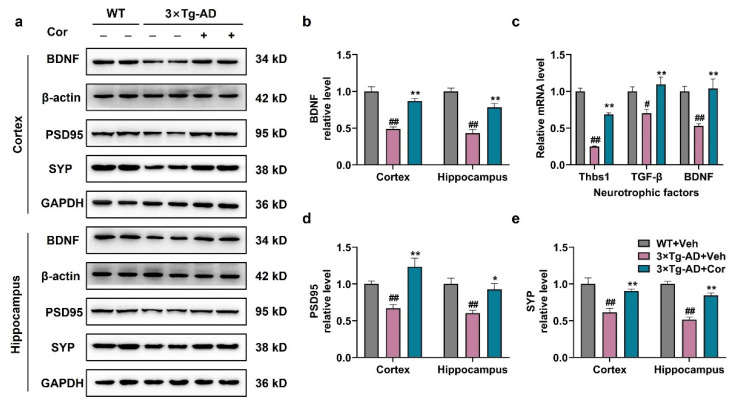
Cornuside increased neurotrophic factors levels and repaired synaptic function in 3 × Tg-AD mice. (**a**) Typical micrographs and (**b**,**d**,**e**) quantitative assessments of hippocampal and cortical BDNF, SYP, and PSD95 proteins for the 3 × Tg-AD mice with or without cornuside therapy versus the WT mice (*n* = 6 per group). (**c**) The mRNA expression of neurotrophic factors including Thbs1, TGF-β, and BDNF were detected in 3 × Tg-AD mice with or without cornuside administration in contrast to the WT controls (*n* = 5 per group). Data are all denoted as mean ± SEM. ^#^ *p* < 0.05 and ^##^ *p* < 0.01 compared with WT + Veh; * *p* < 0.05 and ** *p* < 0.01 compared with 3 × Tg-AD + Veh group.

**Figure 7 nutrients-14-03179-f007:**
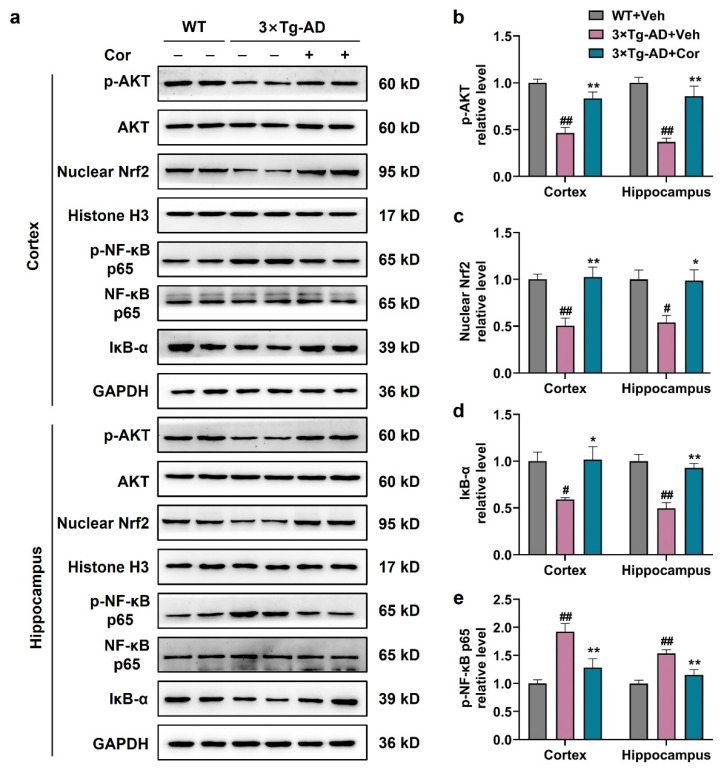
Cornuside activated the AKT/Nrf2 pathway and inhibited NF-κB signaling in 3 × Tg-AD mice. (**a**) Typical WB outcomes and (**b**–**e**) quantitative assessments of hippocampal and cortical p-AKT, nuclear Nrf2, IκB-α, and p-NF-κB p65 proteins for the 3 × Tg-AD mice in the presence or absence of cornuside compared with WT mice. GAPDH or Histone-H3 was used as the control. Data are all expressed as mean ± SEM (*n* = 6 in each group). ^#^ *p* < 0.05 and ^##^ *p* < 0.01 compared with WT + Veh mice; * *p* < 0.05 and ** *p* < 0.01 compared with 3 × Tg-AD + Veh group.

**Figure 8 nutrients-14-03179-f008:**
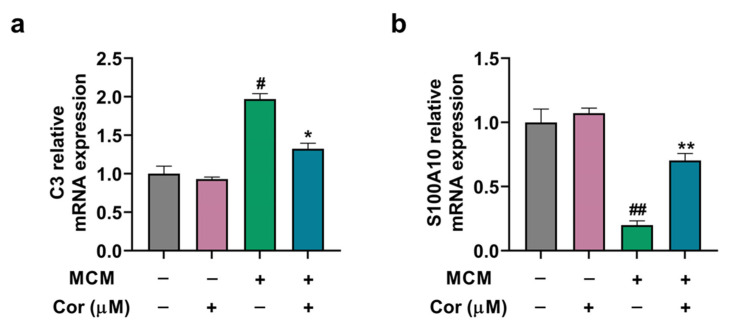
Cornuside regulated A1/A2 phenotypic switch of C6 cells induced by MCM. (**a**) C3 mRNA level and (**b**) S100A10 mRNA level in C6 cells were detected in response to the microglia-conditioned medium or control medium. GAPDH was used as the control. Data are all expressed as mean ± SEM from quadruplicate experiments (*n* = 4 per group). ^#^
*p* < 0.05 and ^##^
*p* < 0.01 compared with MCM (−)/Cor (−); * *p* < 0.05 and ** *p* < 0.01 compared with MCM (+)/Cor (−).

**Figure 9 nutrients-14-03179-f009:**
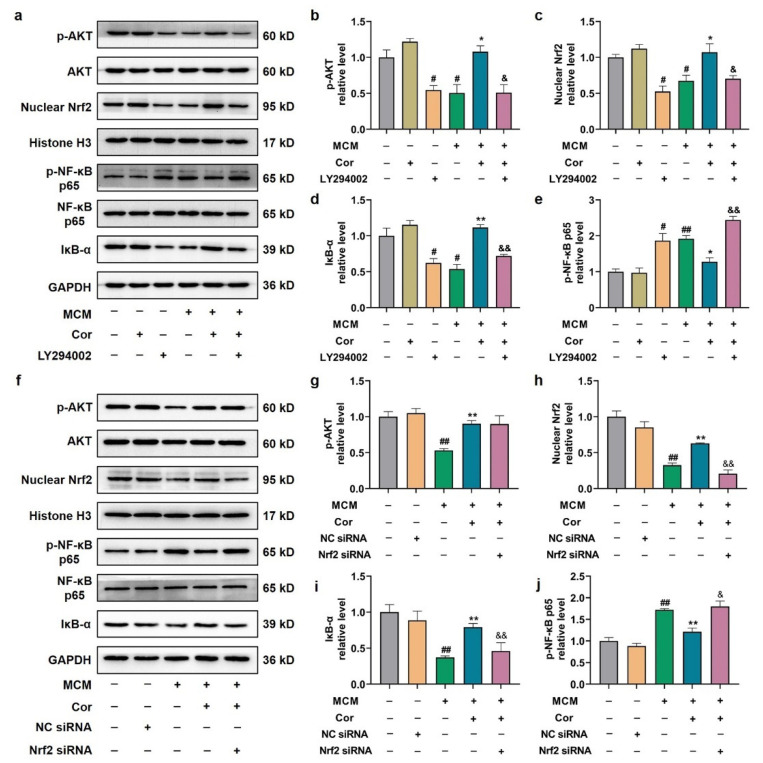
Cornuside protected C6 cells induced by MCM via AKT/Nrf2/NF-κB**.** (**a**) C6 cells were subjected to a 4 h pretreatment with 10 μM of LY294002 and another 24 h treatment with 10 μM of cornuside, and were then exposed to MCM for 24 h. WB was conducted for quantifying the protein levels. (**b**–**e**) Quantification analysis of p-AKT, nuclear Nrf2, IκB-α, and p-NF-κB p65 was conducted in LY294002 or vehicle-treated C6 cells with or without cornuside treatment in response to MCM or control medium. GAPDH or Histone-H3 was used as the control. (**f**) Following a 24 h transfection with NC or Nrf2 siRNA, the C6 cells were subjected to a 24 h treatment with 10 μM of cornuside, and subsequently a 24 h exposure to MCM. WB was performed for the protein level quantification. (**g**–**j**) Quantification analysis of p-AKT, nuclear Nrf2, IκB-α, and p-NF-κB-p65 was conducted in Nrf2 siRNA or scrambled siRNA-treated C6 cells with or without cornuside treatment in response to MCM or control medium. GAPDH or Histone-H3 was used as the control. Data are presented as mean ± SEM, *n* = 3 experiments. (**a**–**e**) ^#^ *p* < 0.05 and ^##^ *p* < 0.01 versus MCM (−)/Cor (−)/LY294002 (−); * *p* < 0.05 and ** *p* < 0.01 versus MCM (+)/Cor (−)/LY294002 (−); ^&^
*p* < 0.05 and ^&&^
*p* < 0.01 versus MCM (+)/Cor (+)/LY294002 (−). (**f**–**j**) ^##^ *p* < 0.01 versus MCM (−)/Cor (−)/NC siRNA (−)/Nrf2 siRNA (−); ** *p* < 0.01 versus MCM (+)/Cor (−)/NC siRNA (−)/Nrf2 siRNA (−); ^&^
*p* < 0.05 and ^&&^
*p* < 0.01 versus MCM (+)/Cor (+)/NC siRNA (−)/Nrf2 siRNA (−).

**Figure 10 nutrients-14-03179-f010:**
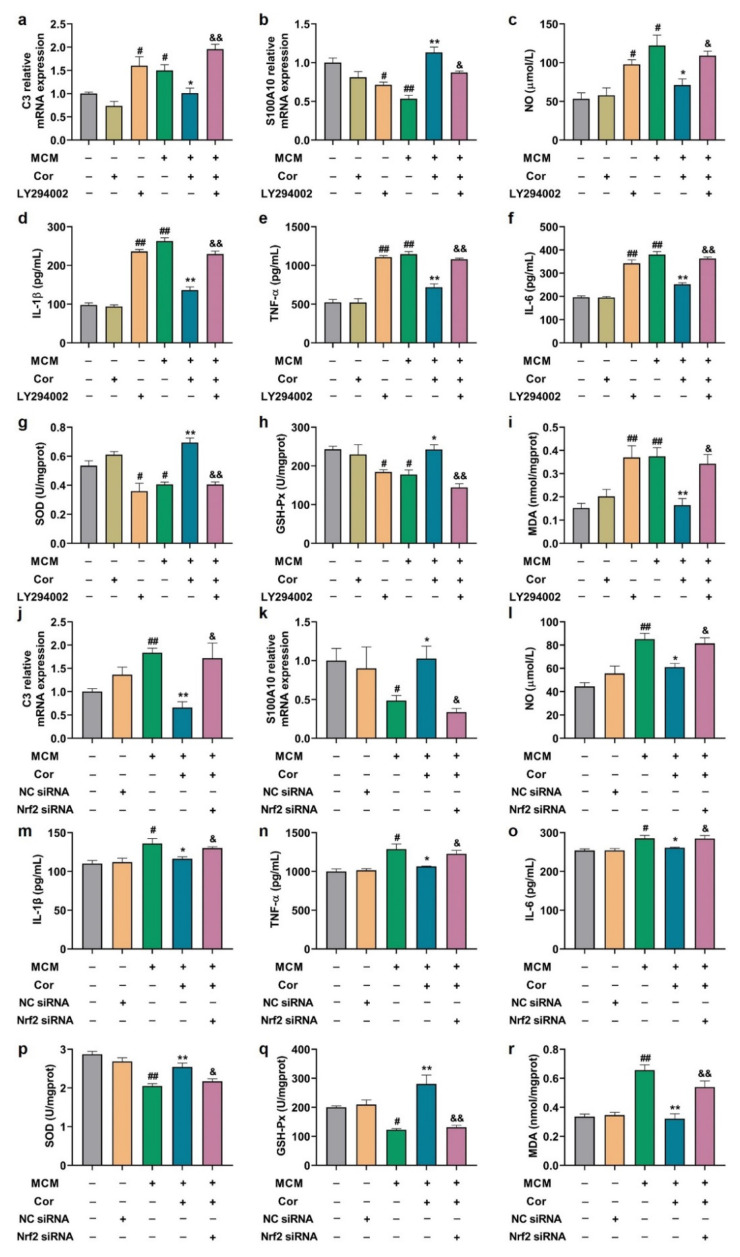
The AKT/Nrf2/NF-κB axis was associated with cornuside-elicited A1/A2 and antioxidase expression and action against neuroinflammation in C6 cells. (**a**–**i**) C6 cells pretreated with LY294002 (10 μM) for 4 h were subjected to a 24 h treatment with 10 μM of cornuside, and subsequently to a 24 h MCM exposure. (**a**,**b**) mRNA levels of A1 marker C3 and A2 marker S100A10 were detected in C6 cells. (**c**–**f**) The concentrations of NO, IL-6, IL-1β, and TNF-α in the gathered supernatants were quantified. (**g**–**i**) The GSH-Px, MDA levels, and the SOD activity were determined in collected supernatant. (**j**–**r**) Following a 24 h transfection with NC or Nrf2 siRNA, the C6 cells were subjected to a 24 h treatment with 10 μM of cornuside, and subsequently to a 24 h MCM exposure. (**j**,**k**) The C3 and S100A10 mRNA levels in C6 cells. (**l**–**o**) The concentrations of NO, IL-6, IL-1β, and TNF-α in the gathered supernatants were quantified. (**p**–**r**) The GSH-Px, MDA expressions, and SOD activity were determined in collected supernatant. Data are presented as mean ± SEM, *n* = 3 experiments. (**a**–**i**) ^#^ *p* < 0.05 and ^##^ *p* < 0.01 versus MCM (−)/Cor (−)/LY294002 (−); * *p* < 0.05 and ** *p* < 0.01 versus MCM (+)/Cor (−)/LY294002 (−); ^&^
*p* < 0.05 and ^&&^
*p* < 0.01 versus MCM (+)/Cor (+)/LY294002 (−). (**j**–**r**) ^#^ *p* < 0.05 and ^##^ *p* < 0.01 versus MCM (−)/Cor (−)/NC siRNA (−)/Nrf2 siRNA (−); * *p* < 0.05 and ** *p* < 0.01 versus MCM (+)/Cor (−)/NC siRNA (−)/Nrf2 siRNA (−); ^&^
*p* < 0.05 and ^&&^
*p* < 0.01 versus MCM (+)/Cor (+)/NC siRNA (−)/Nrf2 siRNA (−).

**Figure 11 nutrients-14-03179-f011:**
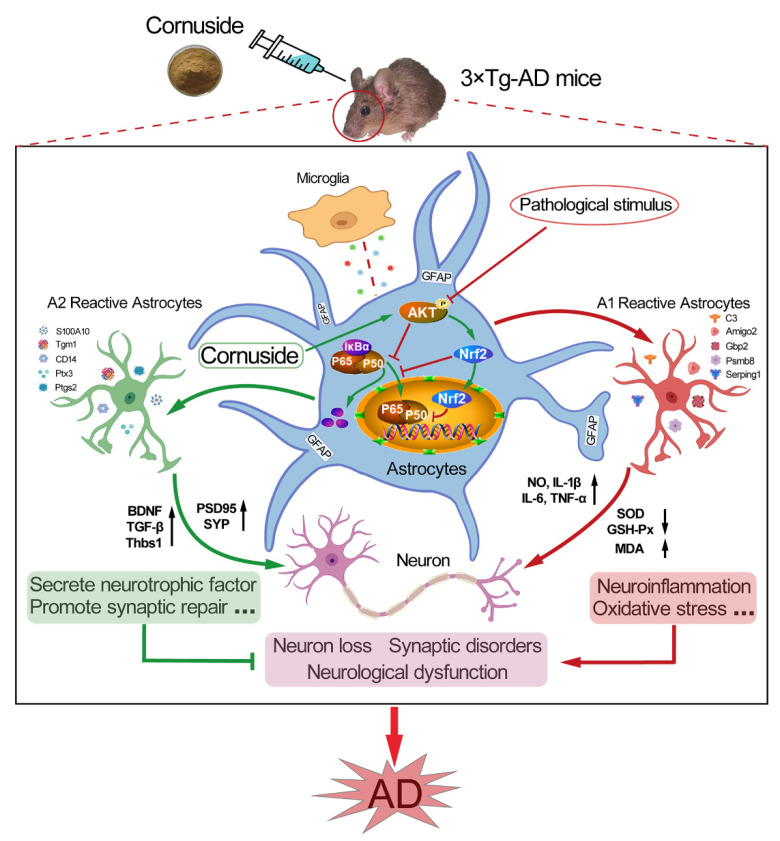
Mechanism of cornuside in improving the cognitive function of the 3 × Tg-AD mice. In the 3 × Tg-AD mice, cornuside regulates the phenotypic conversion of astrocytes, enhances synaptic plasticity, inhibits neuroinflammation and oxidative stress, and ameliorates deficient cognition through the AKT/Nrf2/NF-κB signaling pathway.

## Data Availability

The data supporting the conclusions of this article can be made available from the corresponding author upon request.

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
