# Peer review of "Cornuside Is a Potential Agent against Alzheimer’s Disease via Orchestration of Reactive Astrocytes"

_nutrients, 2022, doi:10.3390/nu14153179_

Round 1

Reviewer 1 Report

The authors suggested that Cornuside has a potential role in prevention and treatment of Alzheimer's Disease.

The experiments and results are several and different but the AD animal perhaps the effect of Ceronuside could be less promising? Are the authors sure that the effect will also be positive in the cells of sporadic subjects?

The major form of AD is the sporadic one. Perhaps the effect of Cornuside could be less promising in sporadic AD? 

What the authors suppose about the amyloid plaques that might be destroy by Ceronuside? The Amilode  oligomers are much more toxic than Amilode plaques,  the authors hypothesized what role these oligomers might play?

Author Response

Point 1: The experiments and results are several and different but the AD animal perhaps the effect of Cornuside could be less promising? Are the authors sure that the effect will also be positive in the cells of sporadic subjects?

The major form of AD is the sporadic one. Perhaps the effect of Cornuside could be less promising in sporadic AD?.

Response 1: Thank you very much for your comment. We agree that the results in animal experiments may be different. In our study, we chose a classic AD animal model. Although this model mainly represents the familial AD model, our results showed that Cornuside had beneficial effects on the main pathological features of AD, including Aβ deposition, glial overactivation, neuroinflammation and tau protein level. We speculate that Cornuside may also play a positive role in sporadic AD, and we will select appropriate animal models and cell models for our further studies.

Point 2: What the authors suppose about the amyloid plaques that might be destroy by Ceronuside? The Amilode oligomers are much more toxic than Amilode plaques, the authors hypothesized what role these oligomers might play?

Response 2: Thanks a lot for your precise comments. The results of our study indicate that Cornuside has a positive regulatory effect on astrocytes in AD mouse model. It is in line with the previous studies that astrocytes and microglia can cluster around and further clear amyloid plaques. According to our data, cornuside can clear amyloid plaques by promoting the transformation of A2 astrocytes.

The amyloid beta (Aβ) cascade hypothesis of Alzheimer’s disease (AD), which was proposed originally by Hardy and Allsop, assumes that imbalance between production and clearance of Aβ in the brain leads to its accumulation, oligomerization, aggregation and formation of Aβ plaques. Oligomers have been indicated as the most toxic Aβ species, appearing likely before plaque deposition at an early stage of AD pathology. Aβ peptide is released from cells in a soluble form, and progressively undergoes aggregation forming oligomers, multimers, and fibrils, ending with deposition of extracellular plaques. These oligomers are thought to cause inflammation, oxidative stress, microglia and astrocyte activation, Tau hyperphosphorylation and apoptosis, there by resulting in synaptic and neuronal loss.

Reviewer 2 Report

The study tested the plant extract, Corni Fructus, the dried fruit of Cornus officinalis Sieb. et Zucc, widely used in Oriental or alternative medicine through intraperitoneal injections for 4 weeks in animals. While the study result appears to support the conclusion drawn, the reviewer has some concerns and questions to be addressed further:

Major questions:

1. What is the tau protein level in your mouse model of AD? You did not assay tau. Rather you only assayed amyloid beta. A-beta hypothesis failed. Why did you keep using a-beta as a marker for AD?

2. You show many abbreviations that are not defined in the abstract. For example, MCM, LPS etc. are not defined in abstract. Please define all abbreviations so that one can understand what you try to describe.

3. Overall, the sample size is very much limited (n=3 to 4 per group). How did you come up with so many figures with statistical significances only with sample sizes of n=3 or 4? Did you conduct power analysis to calculate the appropriate sample size per you group, before you began your study? Any chances for you to be biased against a lack of significance with the limited samples? 

Minor questions:

1. Fig. 2e - atrophy of hippocampus observed: any chances of ventriculomegaly in your animal brains? If so, can you show?

2. Fig. 3c - where cortex? Visual cortex? Frontal cortex? Olfactory cortex? Parietal Cortex? Which part of the cerebral cortex? Please clarify.

3. Fig. 5 - (n = 4 ) Please show a low magnification micrograph so that we can see the big picture of where you are? Which stat test did you use for n = 4? Parametric or non-parametric? Would samples be distributed normally so that you were able to use t test or ANOVA (parametric)? Or you run the Mann-Whitney or Kruskall-Wallis (non-parametric) due to limited samples?

5. Fig. 6 (n = 3 to 4) Wow, there is a significance?

6. Fig. 7 (n = 4) but in Western blot, your n = 2 per group (Fig. 7a). Please show all results if n = 4?

7. Fig. 8 (n = 3) Wow, there is a significance at n = 3?

Author Response

Point 1: What is the tau protein level in your mouse model of AD? You did not assay tau. Rather you only assayed amyloid beta. A-beta hypothesis failed. Why did you keep using a-beta as a marker for AD?

Response 1: Thanks a lot for your constructive suggestion. Deposition of Aβ is still considered to be one of the important pathogenic causes of AD and we focused on the role of astrocytes in AD in this study. Hence, we examined deposition of Aβ and activation of glial cells. As suggested, we performed Western Blot to analyze the effect of cornuside on the expression of p-Tau (S396) and added the result in Figure 2e-f.

Point 2: You show many abbreviations that are not defined in the abstract. For example, MCM, LPS etc. are not defined in abstract. Please define all abbreviations so that one can understand what you try to describe.

Response 2: Thank you for your valuable comment. All the abbreviations have been defined in the abstract according to your suggestion (highlighted in yellow).

Point 3: Overall, the sample size is very much limited (n=3 to 4 per group). How did you come up with so many figures with statistical significances only with sample sizes of n=3 or 4? Did you conduct power analysis to calculate the appropriate sample size per you group, before you began your study? Any chances for you to be biased against a lack of significance with the limited samples?

Response 3: Thank you for your valuable comment. As the 3×Tg-AD model mice we used in this study were well-genotyped and had stable phenotype, the differences among the mouse model were small. In addition, our previous study using the same 3×Tg-AD model was also published (Acta Pharmacol Sin. 2022 Apr;43(4):840-849). Therefore, we could get significant differences from limited sample size (n=3 or 4) based on our previous experience. In this study, we had no biased selection for the samples to be analyzed. However, we do agree that the sample size of 3 to 4 is not sufficient. To further validate our conclusions, we added 2 more samples per group for further analysis. In combination with the previous data, we got the similar results and conclusions. All the data were updated and please check the revisions in Figure 2c-d, Figure 3a, b, d-f, Figure 4a-e, Figure 5, Figure 6, Figure 7.

Minor questions:

Point 1: Fig. 2e - atrophy of hippocampus observed: any chances of ventriculomegaly in your animal brains? If so, can you show?

Response 1: In 3×Tg-AD mice, the hippocampus was atrophied, resulting in ventriculomegaly. Unfortunately, the results of ventriculomegaly cannot be provided at present, which will be further explored in future studies.

Point 2: Fig. 3c - where cortex? Visual cortex? Frontal cortex? Olfactory cortex? Parietal Cortex? Which part of the cerebral cortex? Please clarify.

Response 2: In this study, we only focused on the prefrontal cortex in Figure 3c.

Point 3: Fig. 5 - (n = 4) Please show a low magnification micrograph so that we can see the big picture of where you are? Which stat test did you use for n = 4? Parametric or non-parametric? Would samples be distributed normally so that you were able to use t test or ANOVA (parametric)? Or you run the Mann-Whitney or Kruskall-Wallis (non-parametric) due to limited samples?

Response 3: Thanks a lot for your constructive suggestion. Maybe you are concerned about Fig. 4. We are very sorry that we only considered the co-localization of GFAP and C3 in immunofluorescence double staining and did not take 100× images. Data were expressed as mean ± SEM. One-way ANOVA was employed to make comparisons among groups, and in the case of normal distribution, Tukey’s post hoc test was conducted. Otherwise, the Kruskal–Wallis test was adopted. We added the sample size to n=5 to 6 in animal experiments. In our future studies, we will pay special attention to the sample size, and get the sufficient sample size at the time of study design.

Point 5: Fig. 6 (n = 3 to 4) Wow, there is a significance?

Response 5: As mentioned before, the 3×Tg-AD model mice we used in this study were well-genotyped and had stable phenotype, Therefore, we could get significant differences from limited sample size (n=3 or 4). However, we do agree that the sample size of 3 to 4 is not sufficient. To further validate our conclusions, we added 2 more samples per group for further analysis. All the data were combined, and one-way ANOVA followed by Tukey’s post hoc test was performed for statistical analysis. Finally, we still got the similar results and conclusion. Please check the revisions in Figure 6.

Point 6: Fig. 7 (n = 4) but in Western blot, your n = 2 per group (Fig. 7a). Please show all results if n = 4?

Response 6: Thanks a lot for your kind reminder. According to your suggestion, we added the sample size to n=6 and all results have been shown below. We replaced the previous result in Figure 7.

Point 7: Fig. 8 (n = 3) Wow, there is a significance at n = 3?

Response 7: The cell model we used in this study were stable, Therefore, we could get significant differences from limited sample size (n=3). However, we do agree that the sample size of 3 is not sufficient. To further validate our conclusions, we added an independent experiment for further analysis. In total, 4 samples per group were analyzed. One-way ANOVA followed by Tukey’s post hoc test was performed for statistical analysis by prism8.0. Please check the revisions in Figure 8.

Round 2

Reviewer 2 Report

Authors appear to make a significant revision to the manuscript. Only concern is the sample size (not high), but we will see if the data presented here will be reproduced by others! I believe science is a pursuit of a liberal idea in a most conservative way/method. In that context, authors are responsible for the reproducibility of the work presented here. I hope that all data shown here will be reproducible.